# Low Doses of PFOA Promote Prostate and Breast Cancer Cells Growth through Different Pathways

**DOI:** 10.3390/ijms23147900

**Published:** 2022-07-18

**Authors:** Aurélie Charazac, Charlotte Hinault, Bastien Dolfi, Solène Hautier, Célia Decondé Le Butor, Frédéric Bost, Nicolas Chevalier

**Affiliations:** 1Université Côte d’Azur, INSERM U1065, C3M, Bâtiment Universitaire Archimed, 151 Route de Saint-Antoine de Ginestière, BP 2 3194, CEDEX 3, 06204 Nice, France; aurelie.charazac@gmail.com (A.C.); charlotte.hinault@univ-cotedazur.fr (C.H.); bastien.dolfi@univ-cotedazur.fr (B.D.); solene.hautier@wanadoo.fr (S.H.); celia-deconde@outlook.fr (C.D.L.B.); 2Équipe Labelisée Ligue Nationale Contre le Cancer; 3Université Côte d’Azur, CHU, INSERM U1065, C3M, Hôpital de l’Archet 2, 151 Route de Saint-Antoine de Ginestière, CS 23 079, CEDEX 3, 06202 Nice, France

**Keywords:** endocrine disrupting chemicals, PFOA, hormone-dependent cancers

## Abstract

Endocrine Disrupting Compounds (EDCs) are found in everyday products. Widely distributed throughout the environment, persistent organic pollutants (POPs) are a specific class of EDCs that can accumulate in adipose tissue. Many of them induce adverse effects on human health—such as obesity, fertility disorders and cancers—by perturbing hormone effects. We previously identified many compounds with EDC activity in the circulation of obese patients who underwent bariatric surgery. Herein, we analyzed the effects of four of them (aldrin, BDE28, PFOA and PCB153) on two cancer cell lines of hormone-sensitive organs (prostate and breast). Each cell line was exposed to serial dilutions of EDCs from 10^−6^ M to 10^−12^ M; cytotoxicity and proliferation were monitored using the IncuCyte^®^ technology. We showed that none of these EDCs induce cytotoxicity and that PFOA and PCB153, only at very low doses (10^−12^ M), increase the proliferation of DU145 (prostate cancer) and MCF7 (breast cancer) cells, while the same effects are observed with high concentrations (10^−6^ M) for aldrin or BDE28. Regarding the mechanistic aspects, PFOA uses two different signaling pathways between the two lines (the Akt/mTORC1 and PlexinD1 in MCF7 and DU145, respectively). Thus, our study demonstrates that even at picomolar (10^−12^ M) concentrations PFOA and PCB153 increase the proliferation of prostate and breast cancer cell lines and can be considered possible carcinogens.

## 1. Introduction

Prostate and breast cancers are the most frequent hormone-dependent cancers. With 1.3 and 2.1 million cases each year, respectively, prostate cancer (PCa) is the second most common cancer in men and breast cancer (BCa) the most common in women. They are responsible for approximately 60,000 and 627,000 annual deaths, respectively, and therefore represent a major global health problem [1]. Metastatic relapse is considered the leading source of mortality in these cancers: in some patients, metastasis occurs before first line therapy (including surgery) or can also escape dormancy and growth owing to abnormal angiogenesis [2]. Thus, a better understanding of metastatic progression is crucial to improve patient survival and quality of life.

Several studies have suggested that exposure to endocrine disrupting compounds (EDCs) increases the risk of hormone-dependent cancers [3,4]. In fact, some of them are considered carcinogens by the International Agency for Research on Cancer (IARC), such as 2,3,7,8-tétrachlorodibenzo-p-dioxine (TCDD), some polychlorinated biphenyls (PCBs), dichlorodiphenyltrichloroethane (DTT) for PCa and BCa [5,6] or chlordecone for PCa [7]. Indeed, EDCs can mimic and/or interfere with hormone signaling and intracellular pathways implicated in numerous biological processes including cell proliferation, cell migration, invasion or apoptosis [8]. Furthermore, exposure to environmental doses of EDCs, such as PFOA (perfluorooctanoic acid) (around 200 ng/L), have been reported to affect gene expression in animals and possibly in humans [9], thus underlining the possible carcinogenic activity for these compounds. Among EDCs, persistent organic pollutants (POPs) are lipophilic components that display a relatively long half-life in the human body. Once accumulated in adipocytes, they can be gradually released throughout a lifetime, long after the initial exposure, and then exert delayed adverse effects on organs and tissues [10]. Their release from adipose tissue can be accelerated by a rapid weight loss, as seen in patients who undergo bariatric surgery [11,12] but also in patients with advanced neoplasms exhibiting severe denutrition. As previously reported with bisphenol A, which inhibits cisplatin-induced cytotoxicity in BCa cells [13], we postulate that POP release from peritumoral adipose tissue may also interfere with cancer treatments.

Until now, studies have investigated mostly the role of POPs in carcinogenesis while few reports addressed the role of POPs in cancer aggressiveness and metastasis. Furthermore, most studies have evaluated the carcinogenic potential of high concentrations of POPs reflecting acute and accidental exposures, while biomonitoring of the general population revealed a chronic exposure at low or very low doses. Low-dose effects, as nonmonotonic responses, are remarkably common in studies of natural hormones and EDCs and cannot be predicted by the effects observed at higher doses [14,15]. Thus, the role of POPs in human carcinogenesis needs to be reassessed considering exposure to such low doses.

POPs are usually classified in five main chemical families: (1) the chlorinated family aromatics that include PCBs; (2) dioxins, considered carcinogens by the IARC; (3) perfluorinated compounds; (4) organochlorine pesticides (DTT for example) and (5) flame retardants from the family of polybrominated diphenyl ethers (BDE). In the present study, we analyzed the effects of POPs on human PCa and BCa cell proliferation. These POPs correspond to those that we previously reported in obese patients that underwent bariatric surgery [11]. Hence, we explored the effects of aldrin (an organochloride insecticide now banned in most countries, but which remains a matter of concern in some French areas), BDE28 (a polybrominated flame-retardant found in many electronic components wastes and furniture), and PCB153 and PFOA (both already associated with carcinogenic effects) [16,17,18]. We chose to test a wide range of doses from 10^−6^ to 10^−12^ M, in order to decipher potential non-monotonic effects around usual plasma concentrations of these POPs [19,20]. We showed that among the four tested POPs, only PFOA and PCB153 induce cancer cell proliferation at very low concentration (10^−12^ M). Indeed, in both PCa and BCa cells, this proliferative effect is limited to very low concentrations, whereas high doses (10^−6^ M) had no effect. Moreover, we showed that PFOA promotes growth of both cell lines through different pathways.

## 2. Results

### 2.1. Picomolar Concentrations of PFOA and PCB153 Increase Prostate and Breast Cancer Cell Proliferation

PCa (DU145) and BCa (MCF7) cell lines were exposed to serial dilutions (from 10*^−^*^6^ M to 10*^−^*^12^ M) of aldrin, BDE28, PCB153 or PFOA, and cell proliferation was monitored by real time imaging (Figure 1). Using a cytotoxicity assay, we showed that POPs did not induce a significant cytotoxic effect on both cell lines even at the higher concentrations (Appendix A). As expected, Hepatocyte Growth Factor (HGF) and Estrogen (E2) significantly increased cancer cell proliferation for, respectively, DU145 and MCF7 (Figure 1A–D) [21,22].

For DU145 cells, aldrin, BDE, PCB153 and PFOA significantly increased cell proliferation (Figure 1A,C). While this proliferative effect was limited to high and low concentrations for aldrin and BDE (10*^−^*^6^ M and 10*^−^*^9^ M), we observed an increased proliferation for PCB153 and PFOA treatments (30% and 23%) at very low (picomolar) concentration (10*^−^*^12^ M). Unlike PCa cells, aldrin (10^−12^ M) and BDE28 (10*^−^*^6^ M) induced a decrease in BCa cell proliferation, while picomolar concentrations of PCB153 and PFOA significantly increased MCF7 cell growth by 25% and 32%, respectively (Figure 1B,D). Interestingly, the same chemicals at higher concentrations (10*^−^*^9^ M and 10*^−^*^6^ M) exhibited no effect on cell proliferation in this assay in both cell lines. 

Because we observed the same proliferative effect of PCB153 and PFOA at very low concentrations in both PCa and BCa cell lines, we focused our attention on those two chemicals (Figure 2A,B). We performed a BrDU proliferation assay comparing the effects of 10*^−^*^9^ M and 10*^−^*^12^ M concentrations of PCB153 and PFOA to the well-known proliferative effects of HGF (DU145) and E2 (MCF7) (Figure 2C,D). We confirmed that both concentrations of PCB153 promoted cell growth in DU145 and MCF7, while PFOA increased cell proliferation by 30% and 31% in DU145 and MCF7, respectively, only at the lowest concentration of 10*^−^*^12^ M. 

### 2.2. The Proliferation Induced by PFOA Is Mediated by Plexin D1 in Prostate Cancer Cells

As only PFOA showed specific proliferative effect at low doses in prostate cancer cells, we chose to focus our experiments on this chemical and to investigate molecular mechanism aspects first in DU145 cells. By using phospho-kinase assays, we observed no modification in the phosphorylation of kinases classically implicated in the proliferation of cancer cells such as Akt, GSK3, mTORC1 or STAT in DU145 cells treated with 10*^−^*^12^ M PFOA (Figure 3A,B). Similarly, we did not observe any significant modification with PCB153 (Appendix A).

We then performed a microarray analysis in DU145 cells exposed to 10*^−^*^12^ M PFOA. After a differential expression analysis to identify significantly dysregulated genes, we used a computational method to explore the molecular signature of gene sets. In the top 10 identified gene sets, 3 were related to cancer. We found that genes related to the NOTCH3 signaling pathway and matrisome were modulated by PFOA (Figure 3C). Among these genes, PlexinD1 (PLXND1), a pivotal mediator of the NOTCH signaling pathway in PCa [23], was significantly upregulated (Figure 3D). In addition, PLXND1 overexpression is correlated with significant lower disease-free survival in PCa patients (Figure 3E). To validate the microarray experiment, we analyzed the expression of PLXND1 mRNA by Q-PCR and showed that PLXND1 is significantly increased after 48 h exposure to PFOA (Figure 3F). Next, we knocked down PLXND1 using a siRNA strategy. In DU145 cells, PLXND1 depletion (50%) (Figure 3F) did not affect the basal proliferation of DU145 cells but it abolished the proliferative effects by PFOA (Figure 3G). These data indicated that PLXND1 is involved in PFOA-induced cell growth in the DU145 cell line. 

### 2.3. Breast Cancer Cells Proliferation Induced by Picomolar Concentrations of PFOA Is Dependent on the PI3K/Akt and mTORC1 Pathways

To identify the signaling pathways modulated by PFOA in MCF7 cells, we also performed a phospho-kinase assay. Low doses of PFOA (10*^−^*^12^ M) increased the phosphorylation of GSK-3α/β on serine 21/9, Akt on threonine 308 and serine 473 as well as p70S6kinase on threonine 389 (Figure 4A,B). In contrast, PCB153 did not affect the phosphorylation of the proteins studied on the array (Appendix A). To confirm these modifications, we performed an immunoblot and showed that the phosphorylation of the three kinases was significantly increased after PFOA exposure in MCF7 cells (Figure 4C,D). The most significant change that we observed was a doubling of P-Akt (Ser473) (Figure 4D). Collectively, these results showed that picomolar concentrations of PFOA induce the phosphorylation of protein kinases of PI3K/Akt/mTORC1 signaling pathways known to play an important role in BCa cell proliferation.

To determine whether the pro-proliferative effect of PFOA was dependent on Akt or p70S6K phosphorylation, we treated MCF7 cells with rapamycin or LY294002, which are potent inhibitors of mTORC1 (an upstream kinase p70S6 kinase), and PI3K, respectively. Rapamycin decreased by 50% the phosphorylation of p70S6K (Figure 5A,C) and abolished PFOA-induced proliferation at a concentration of 40 nM (Figure 5E). LY294002 at 10 µM or 25 µM decreased the phosphorylation of Akt on Ser473 and to a lesser extent of Thr308 in cells treated with PFOA (Figure 5B,D and Appendix A). Interestingly, these concentrations of LY294002 suppressed the proliferative effects of PFOA (Figure 5F). Thus, our results suggest that the proliferation induced by PFOA in MCF7 cells is mediated by the Akt/mTORC1 axis.

## 3. Discussion

Currently, there is a growing social concern about the adverse health effects of pollutants [24]. Some of them have already been classified as carcinogens by the IARC but few studies examine the effects of POPs on established human cancer cells, especially at very low concentrations, which correspond to the chronic exposure of the population [25]. One possible and important feature of the cellular response to chemicals is a non-monotonic dose–response with an inverted U-shape curve. In these conditions, it is relevant to study low doses (nano to picomolar) in experimental studies. In epidemiological studies, high-dose exposure can fail to reveal any association because the exposure range could be in the saturated part of the dose–response [26]. This may explain why studies with low-dose exposure ranges showed much stronger associations than those which examined the exposure to 10 to 100 times higher concentrations of chemicals [27]. The major finding of our study is that picomolar (10^−12^ M) concentrations of PFOA and PCB153 increase the proliferation of PCa and BCa cell lines, which are the most frequent hormone-dependent cancers [1]. 

The pro-proliferative effects of PFOA are associated with important intracellular modifications implicating the Semaphorin-PLXDN1 and the Akt/mTORC1 signaling pathways that are frequently activated in both cancers [28]. Perfluoroalkyl substances in tap water has been associated with a higher risk of prostate cancer [25], while the presence of low serum concentrations of PFOS and PFOA are associated with endocrine receptor-negative breast cancer tumors [29]. Nevertheless, few studies have investigated the effects of PFOA on cancer cells in vitro. PFOA has been shown to stimulate colorectal and ovarian cancer cell invasion at nanomolar (10^−9^ M) concentrations through the upregulation of metalloproteases and activation of NF-κB but did not affect cancer cell growth [30,31]. Similar positive effects on cell migration were obtained in the BCa cell line MDA-MB-231, while cell growth was not affected [32]. In accordance with this study, we did not observe any effect on proliferation at 10^−9^ M rather only at picomolar (10^−12^ M) concentrations. In the body, hormones are active at extremely low doses and below the picomolar (10^−12^ M) range, but it is also known that low (nanomolar [10^−9^ M]) concentrations of EDCs induce biological changes such as modification of miRNA expression in mice testes [33]. Importantly, the concept that low doses of EDCs could have a biological effect start to be well-accepted by scientists [34]. In accordance with our data, the team of Gael Prins recently showed that per- and polyfluorinated alkyl substances (PFAS) at nanomolar (10^−9^ M) concentrations increase the size and the number of prostaspheres and enhance tumor growth in a xenograft model [35]. This is a main concern for regulatory agencies since these very low concentrations correspond to those to which we are routinely exposed. Further experiments are also requested to investigate the effects of these compounds to induce cancer promotion in normal prostate and breast cell lines.

We showed that PLXND1 is transcriptionally upregulated by PFOA. To our knowledge this is the first time that EDCs are shown to regulate an actor of the PLXND1/NOTCH signaling pathway. NOTCH signaling has been shown to upregulate the PLXND1 promoter activity in PCa cells [23], but VEGF is also a regulator of PLXND1 [36]. In addition, Semaphorins 3A and 2E, the ligands of PLXND1, are known to be overexpressed in PCa [37]. However, PFOA has never been described to regulate these molecular entities. Future studies will help to decipher the mechanism implicated in the regulation of PLXDN1 expression by PFOA. Interestingly, in accordance with that, analysis of the TCGA-PRAD database showed that PLXND1 overexpression is significantly associated with a poor disease-free survival in PCa, suggesting that EDCs could participate in cancer aggressiveness (and metastasis) through this new signaling pathway.

It is well-known that PFOA is also able to bind to the nuclear receptor Peroxisome proliferator activated receptor-alpha (PPARα) and thus alters lipid metabolism and induces liver toxicity in rodents [38]. Our data demonstrate that in MCF7 and DU145 cell lines, the activity of PPARα is not altered by PFOA treatment (data not shown). Interestingly, Hu et al. demonstrated that PFAS do not alter PPARα expression in prostate stem and progenitor cells [35]. Therefore, we can hypothesize that PFAS and PFOA can use other signaling pathways than the one of PPAR in PCa pathogenesis.

It was demonstrated in liver and muscle that PFOA induces the phosphorylation of Akt on Ser473 through the downregulation of Phosphatase and TENsin homolog (PTEN) [39]. In human Ishikawa endometrial cancer cells, mTORC1 is activated and mediates, along with the ERK pathway, the pro-migratory and invasive effects of PFOA [40]. In accordance with these studies, we showed that PFOA phosphorylates Akt and activates mTORC1 in MCF7 cells. The Akt/PI3K and mTORC1 pathways are frequently activated in breast cancer cells and therapies targeting these pathways belong to the current therapeutic arsenal of the disease [41]. Interestingly, we showed that both pathways mediate the proliferative effects of PFOA in breast cancer cells; further investigations are needed to determine the mechanism linking PFOA and the intracellular machinery in cancer cells as we can suppose that PFOA could interfere with these targeted therapies like previously reported for bisphenol A cells [13].

PFOA is used in several industrial applications, especially as a surfactant in daily cooking products and food wrappers. Its stability is a matter of environmental concern as it is resistant to degradation by natural processes. Its half-life is estimated to about three years in humans. Thus, despite its being included in the Stockholm Convention in 2015 and its production banned in the US and Europe in the last two years, significant environmental contamination still exists, and we assume, like others, that PFOA can bioaccumulate in the human body. Apart from some occupational areas where people are exposed daily to high doses of PFOA, the usual blood concentrations of PFOA remain low and are quite similar to the concentrations that we used in our in vitro experiments with cancer cell lines [11]. Our results, coupled with epidemiological studies, point out the danger of this substance even at such very low doses. Future research investigating the mode of action of PFOA (and also other PFAS) in more complex models is needed to understand the involvement of these substances in cancer and human pathologies [42].

## 4. Materials and Methods

### 4.1. Material

Aldrin, BDE28, PCB153 and PFOA were purchased from Sigma^®^ (Darmstadt, Germany) and then diluted in ethanol to a 10^−3^ M concentration stock, except for PCB153 which was diluted to a 10^−5^ M concentration stock. LY294002, a PI3K inhibitor, and rapamycin, a mTOR inhibitor, were, respectively, purchased from InVivogen^®^ (Toulouse, France) and Sigma^®^.

### 4.2. Cell Culture

Human BCa cell line (MCF7) and human PCa cell line (DU145) were obtained from ATCC and cultured in Dulbecco’s Modified Eagle’s Medium (DMEM) supplemented with 10% fetal bovine serum (FBS) and 1% penicillin/streptomycin (respectively, 100 U/mL and 100 µg/mL, Gibco^®^ [Asnières sur Seine, France]) at 37 °C in a humidified atmosphere (5% CO_2_). All cell lines used in this study were tested to confirm that they were free of mycoplasma [43].

Treatments with chemicals were done the day after cell seeding in DMEM without red phenol supplemented with 10% heat inactivated and charcoal stripped FBS and 1% penicillin/streptomycin (respectively, 100 U/mL and 100 µg/mL, Gibco^®^).

### 4.3. Transfection

For lentiviral transductions, MCF7 and DU145 cells were transfected with IncuCyte^®^ NucLight Red Lentivirus (Essen Bioscience^®^ [Ann Arbor, MI, USA]) encoding for red fluorescent protein (RFP) in nucleus. Briefly, 4.000 cells/well were seeded in a 96-well plate. Cells were infected according to the manufacturer’s protocol using 8 μg/mL hexadimethrine bromide (Sigma^®^). A total of 24 h after the infection, cells were selected using puromycin (1 µg/mL, Sigma^®^) for seven days.

For siRNA transfections, as previously described [43], siRNA control or siRNA pool directed against PlexinD1 (On Target Plus, Dharmacon^®^ [Lafayette, CL, USA]) were transfected with Lipofectamine RNAiMAX (InVitrogen^®^ [Asnières sur Seine, France]) according to the manufacturer’s instructions. Briefly, 120.000 cells/well in a 6-well plate were seeded overnight. Mix of OptiMEM (Gibco^®^), Lipofectamine RNAiMAX and siRNAs were incubated for 20 min at room temperature prior addition on cells. Cells were then incubated for 72 h.

### 4.4. IncuCyte^®^ Assay

Cells infected with NucLight Red Lentivirus were seeded in a 48-well plate at a concentration of 20.000 cells/well. The day after, chemicals were added in media containing IncuCyte^®^ Cytotox Green Reagent (Essen Bioscience^®^) for loss of cell membrane integrity assessment according to the manufacturer’s protocol [44]. Immediately after, the plate was put on the IncuCyte^®^ (Essen Bioscience^®^ [Ann Arbor, MI, USA]). Three wells were used per replicate and nine pictures per well were taken every 2 h using a 10× lens for each experiment. IncuCyte^®^ software is used to analyze images for both red nucleus count and green cell dying count. 

### 4.5. BrdU Cell Proliferation Assay

BrdU cell proliferation assay was performed using the Cell proliferation ELISA colorimetric kit (Sigma^®^) according to the manufacturer’s instructions. Briefly [43], cells were seeded overnight in 96-well plate at 4.000 cells/well. The day after, chemical treatments or siRNA transfections were performed. Inhibitors and 5-bromo-2′-deoxyuridine were added overnight prior the end of the experiment. We then followed the manufacturer’s protocol. 

### 4.6. Human Phospho-Kinase Array

Human Phospho-Kinase Antibody Array (RD Systems^®^ [Minneapolis, Minnesota]) was used to analyze the relative site-specific phosphorylation of 43 kinases and 2 related total proteins. Following the manufacturer’s instructions, we used 1.107 cells/mL for cell lysis corresponding to 400 µg of protein per array [43]. Phosphorylated proteins were detected using Syngene PXi by exposing membranes to luminescent signals generated after incubation of the membrane with ECL (WBKLS0500, Millipore^®^ [Darmstadt, Germany]). Image J^®^ software (National Institutes of Health and Laboratory for Optical and Computational Instrumentation [University of Wisconsin, Madison, WI, USA]) was used to quantify dot intensity.

### 4.7. Western Blot

Total protein was extracted using RIPA buffer (Tris 50 mM, NaCl 150 mM, SDS 0.1%, Na Deoxycholate 0.5%, NaF 5 mM, NP40 1%, EDTA 10 mM) plus protease-inhibitor cocktail (Mini-complete, EDTA-free, Roche^®^ [Basel, Switzerland]). After BCA measurement (Interchim^®^ [San Pedro, CA, USA]), samples were heat denatured with 1× Laemmli buffer. Forty micrograms of protein per well were run on 6.5% or 9.5% SDS-PAGE electrophoresis gel following by transfer in PVDF membranes (IPVH00010, Millipore^®^). After membrane blocking in 2% BSA in TBS/0,1% Tween20 for one-hour, primary antibodies (Appendix A) were applied overnight at 4 °C. Then, after washing, membrane was probed one hour at room temperature with the secondary antibody (HRP-conjugated AffiniPure from Jackson ImmunoResearch Laboratories^®^ [Cambridge, UK]). Blots were visualized using Syngene PXi by exposing membranes to luminescent signals generated after incubation of the membranes with ECL (WBKLS0500, Millipore^®^). Image J^®^ software was used to quantify band intensity.

### 4.8. Microarray Experiment

Pangenomic microarrays were performed on the UCA GENOMIX platform (IPMC, CNRS U7275, Sophia-Antipolis, France) using Agilent-072363 SurePrint G3 Human GE v3 8 × 60 K Microarray. RNA from DU145 cells were treated with EtOH (control), 10^−6^, 10^−9^ and 10^−12^ M PFOA for 48 h before microarray analysis. Experimental data and associated microarray designs have been deposited in the NBCI Gene Expression Omnibus (GEO under series GSE150655 and platform record GPL 1456). We then focused on coding sequences to perform a differential expression analysis between EtOH (control) and 10^−12^ M PFOA conditions using Limma from Phantasus software (https://artyomovlab.wustl.edu/phantasus/ accessed on 20 October 2021), and an overlap analysis with significantly deregulated genes was performed using GSEA (Gene Set Enrichment Analysis) software to investigate canonical pathway and reactome gene sets with FDR q-value less than 0.05 (https://www.gsea-msigdb.org/gsea/msigdb/annotate.jsp accessed on 15 November 2021).

### 4.9. RNA Extraction and RT-qPCR

Total RNA was extracted from cell lines using Trizol reagent according to the manufacturer’s instructions (Invitrogen^®^). The quantity and quality of the isolated RNA were checked using NanoDrop™ One Spectrophotometer (Thermo Scientific^®^ [Asnières sur Seine, France]). Reverse transcription of 1 mg of total RNA into first strand cDNA was done using the reverse transcription system (A3500, Promega^®^ [Charbonnières-les-Bains, France]). Fast SYBR Green master mix (Applied Biosystems^®^ [Asnières sur Seine, France]) was used to perform the real-time quantitative PCR on a StepOnePlus system (Applied Biosystems^®^). The relative expression was calculated using the ∆Ct method and RPLP0 mRNA level was used as the endogenous control to normalize relative expression values of each target genes. Gene-specific primer sequences are summarized in the Appendix A.

### 4.10. Statistical Analyses

All experiments were performed in triplicate with 3 wells per condition in each one. Significant differences between two conditions were assessed by unpaired or paired Student’s t-test and two-way ANOVA test followed by post hoc Fisher test when more than two conditions were analyzed [43]. A *p*-value < 0.05 was considered statistically significant. All bars shown represent mean +/− Standard Error Mean (SEM). 

## Figures and Tables

**Figure 1 ijms-23-07900-f001:**
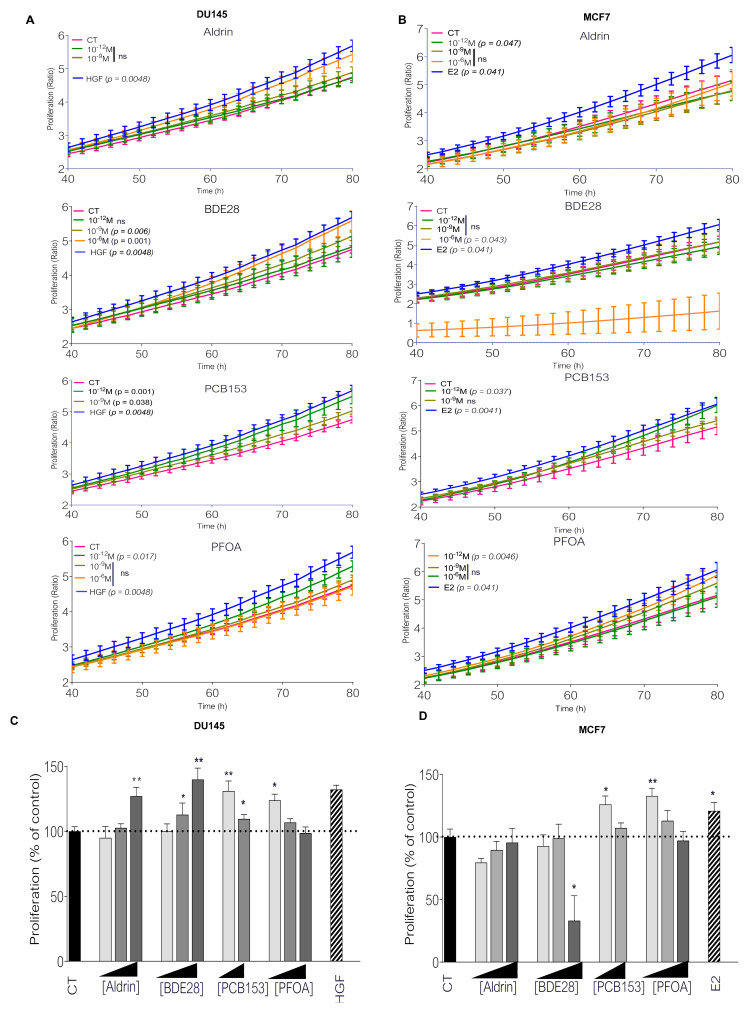
Persistent organic pollutants affect MCF7 and DU145 cell proliferation. (**A**,**B**) IncuCyte^®^ proliferation assay in DU145 and MCF7 cells infected with NucLight red lentivirus, after treatment of 80 h with either aldrin (10^−12^ M, 10^−9^ M or 10^−6^ M), BDE28 (10^−12^ M, 10^−9^ M or 10^−6^ M), PCB153 (10^−12^ M or 10^−9^ M) or PFOA (10^−12^ M, 10^−9^ M or 10^−6^ M) in comparison with cells treated with EtOH (CT). Each point represents a mean ± SEM of 3 independent experiments. (**C**,**D**) IncuCyte^®^ proliferation assay. Slope between 48 h and 72 h of simple linear regression line of IncuCyte^®^ proliferation curve of MCF7 and DU145 cells after treatment with either aldrin (10^−12^ M, 10^−9^ M or 10^−6^ M), BDE28 (10^−12^ M, 10^−9^ M or 10^−6^ M), PCB153 (10^−12^ M or 10^−9^ M) or PFOA (10^−12^ M, 10^−9^ M or 10^−6^ M) in comparison with cells treated with EtOH (CT). Estradiol (E2, 10^−9^ M) and HGF (10 ng/mL) were used as positive control for MCF7 or DU145 cells, respectively. Each bar represents a mean ± SEM of 3 independent experiments. T test. * *p* < 0.05; ** *p* < 0.005.

**Figure 2 ijms-23-07900-f002:**
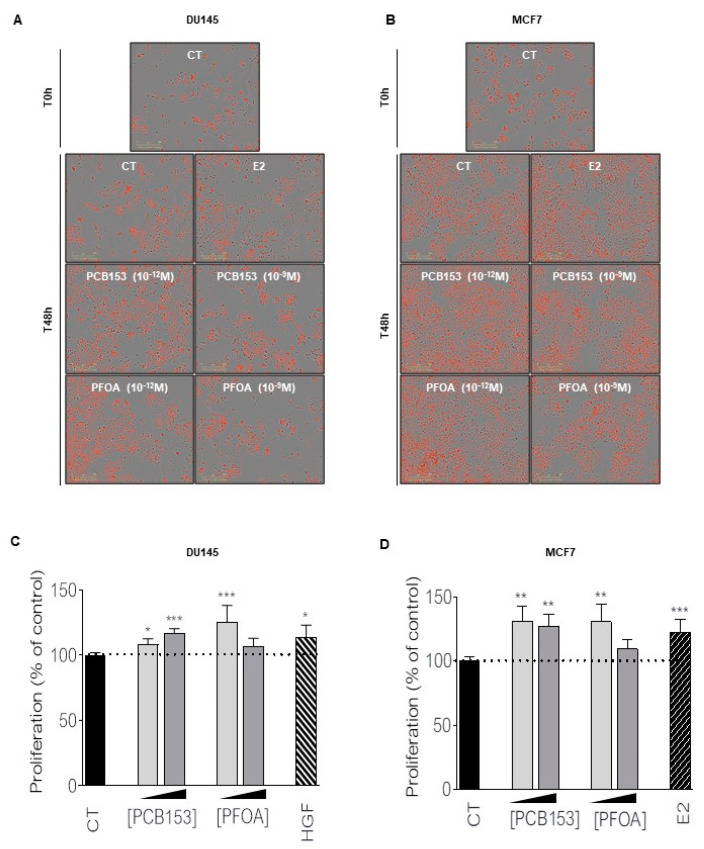
PCB153 and PFOA increase MCF7 and DU145 cells proliferation. (**A**,**B**) DU145 and MCF7 cells infected with NucLight red lentivirus representation of IncuCyte^®^ proliferation assay at T 0 h and T 48 h after treatment of either EtOH (CT), PCB153 (10^−12^ M or 10^−9^ M) or PFOA (10^−12^ M or 10^−9^ M). Estradiol (E2, 10^−9^ M) and HGF (10 ng/mL) were used as positive controls for MCF7 or DU145 cells, respectively. Magnification ×20 (**C**,**D**) BrdU cell proliferation assay. Measured proliferation of MCF7 or DU145 cells after 72 h treatment with either PCB153 (10^−12^ M or 10^−9^ M) or PFOA (10^−12^ M or 10^−9^ M) in comparison with cells treated with EtOH (CT). Estradiol (E2, 10^−9^ M) or HGF (10 ng/mL) were used as positive controls for MCF7 or DU145 cells, respectively. Each bar represents a mean ± SEM of 3 independent experiments. T test. * *p* < 0.05; ** *p* < 0.005; *** *p* < 0.0005.

**Figure 3 ijms-23-07900-f003:**
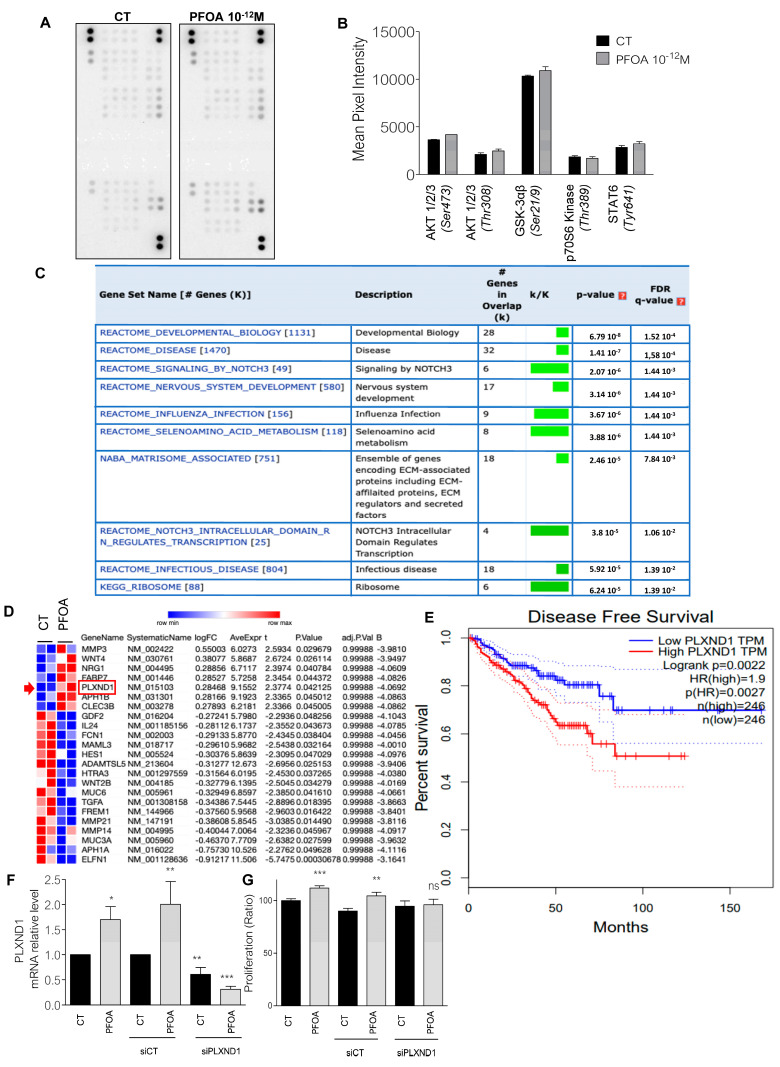
Plexin D1 plays an important role in PFOA-induced proliferation in DU145 cells. (**A**,**B**) Image and quantification of Human Phospho-Kinase Array. Relative levels of protein phosphorylation of a range of 43 kinase phosphorylation sites of DU145 cells after 48 h of treatment with PFOA (10^−12^ M) in comparison with cells exposed with EtOH (CT). (**C**) Top ten gene sets of enrichment analysis from a microarray performed on RNA from cells treated 48 h with 10^−12^ M PFOA. (**D**) Heatmap analysis of genes modified by the treatment with PFOA (10^−12^ M) belonging to Notch3 and matrisome gene sets. (**E**) Disease free survival related to PlexinD1 expression in prostate cancer patients of the TCGA database. (**F**) Expression of PLXND1 mRNA in DU145 cells treated 48 h with 10^−12^ M PFOA. Cells were transfected with siRNA Control (siCT) or siRNA PLXND1. (**G**) Proliferation assay (BrdU incorporation) performed in DU145 cells treated 48 h with 10^−12^ M PFOA, transfected with siCT or siPLXND1. T test. * *p* < 0.05; ** *p* < 0.005; *** *p* < 0.0005; ns: non-significant.

**Figure 4 ijms-23-07900-f004:**
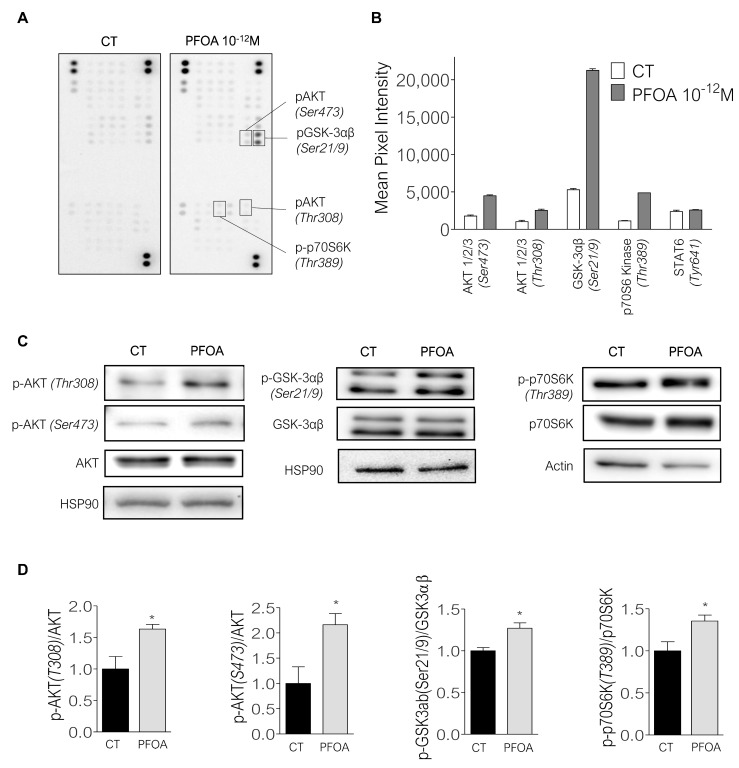
PFOA increases the phosphorylation of Akt and p70S6 kinase in MCF7 cells. (**A**,**B**) Image and quantification of Human Phospho-Kinase Array. Relative levels of protein phosphorylation of a range of 43 kinase phosphorylation sites of MCF7 cells after 48 h of treatment with PFOA (10^−12^ M) in comparison with cells exposed with EtOH (CT). (**C**) Western blot of phospho-AKT (Thr308 and Ser473), AKT total, phopsho-GSK3αβ (Ser21/9), GSK3αβ total, HSP90, phospho-p70S6Kinase (Thr389), p70S6Kinase total and actin of MCF7 cells protein extract after 48 h of treatment with PFOA (10^−12^ M) compared to cells treated with EtOH (CT). (**D**) Quantification of protein level of three independent western blot experiments. T test. * *p* < 0.05.

**Figure 5 ijms-23-07900-f005:**
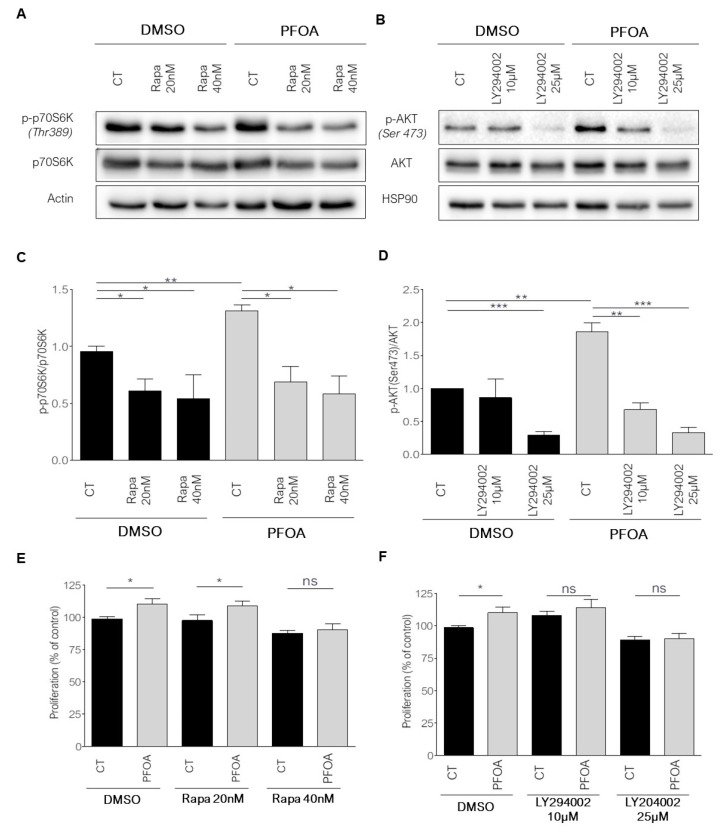
PFOA increases proliferation via Akt and mTORC1 in MCF7 cells. (**A**) Western blot of phospho-p70S6Kinase (Thr389), total p70S6Kinase and Actin of MCF7 cell protein extract after 48 h of treatment with PFOA (10^−12^ M) in comparison with cells treated with EtOH (CT) and overnight exposure with either DMSO or Rapamycin (20 nM or 40 nM). (**B**) Quantification of protein level of three different western blot experiments. (**C**) BrdU cell proliferation assay. Measured proliferation of MCF7 cells after 48 h treatment with PFOA (10^−12^ M) in comparison with cells treated with EtOH (CT) and overnight exposure with either DMSO or Rapamycin (20 nM or 40 nM). Each bar represents a mean ± SEM of 3 independent experiments. (**D**) Western blot of phospho-AKT(Ser473), AKT total and HSP90 of MCF7 protein extract after 48 h of treatment with PFOA (10^−12^ M) in comparison with cells treated with EtOH (CT) and overnight exposure with either DMSO or LY294002 (10 µM or 25 µM) overnight exposure. (**E**) Quantification of protein level of three independent western blot experiments. (**F**) BrdU cell proliferation assay. Measured proliferation of MCF7 cells after 48 h treatment with PFOA (10^−12^ M) in comparison with cells treated with EtOH (CT) and overnight exposure with either DMSO or LY294002 (10 µM or 25 µM). Each bar represents a mean ± SEM of 3 independent experiments. * *p* < 0.05; ** *p* < 0.005; *** *p* < 0.0005; ns: non-significant.

## Data Availability

The data presented in this study are available on request from the corresponding author.

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
