# Peer review of "Low Doses of PFOA Promote Prostate and Breast Cancer Cells Growth through Different Pathways"

_ijms, 2022, doi:10.3390/ijms23147900_

Round 1
Reviewer 1 Report
The manuscript (ijms-1817936) entitled “Low doses of PFOA promote prostate and breast cancer cells growth through different pathways” by Dr. Aurélie and co-workers, reports on a study aimed in assessing the carcinogenic effects of four endocrine disruptors compounds also known as persistent organic pollutants (POPs), including aldrin, BDE28, PFOA and PCB153 prostate and breast cancer cells. Main results indicate that at picomolar (10−12M) concentrations PFOA and PCB153 increase the proliferation of prostate and breast cancer cell lines. Therefore, a carcinogenic role cannot be excluded for these two compounds. In my opinion, the manuscript is in general were organized, while the experimental design is well performed. Data are also well presented, in terms of readability and figures, while also being well discussed. The manuscript is in general well written, clear, and easily to understand. The work will increase our knowledge on the carcinogenic function of POPs, especially in the context of prostate and breast cancer. It therefore will have an adequate impact in these fields. My final recommendation is therefore a minor revision. I have several comments for improving the quality of the manuscript:
1. Line 19 better “Herein, We analyzed…”
2. Lines 42-57 and/or lines 227-232, Exposure to environmental doses of endocrine disruptors compounds, such as PFOA (around 200 ng/L), have been reported to affect gene expression in animals and possibly in humans (PMID: 29105837), thus underling the possible carcinogenic activity for these compounds. This important notion and supporting reference should be included
3. The chemical families of aldrin, BDE28, PFOA and PCB153 should be detailed in the introduction. It should be helpful for the reader
4. The authors should explain why they selected this specific concentration range, i.e., from 10−6M to to 10−12M for all compounds. Previous studies?
5. I would include hours or days in the x axis of figure 1, panel A (all graphs)
6. Figure 3 panels C and E, panels are almost unreadable
7. The rationale behind the selection of these two different tumor models for evaluating the carcinogenic function of POPs should be included in the discussion
8. The transformative/ pro-proliferative effected of these compounds should also be evaluated in normal breast and prostate cell lines, as normal cell model . This can be considered a limitation of the study
9. Several works in the field should be included, such as PMID: 20423814, PMID: 23308854 and PMID: 33385391
10. Supporting references should be included in the methods section, including statistical analyses, as being completely lacking in supporting references
Author Response
We thank Reviewer #1 for considering our work. As expected, we modified our manuscript according to his precious comments:
- we added the term "Herein" at the beginning of the sentence
- we completed our introduction on the carcinogenic effects of PFOA and added the reference to the article of Rotondo et al.
- as suggested, we detailed more the chemical families of aldrin, BDE28 and PCB153, including some references on their potential carcinogenic effects
- we explained in the text why we chose this range of doses: we want to investigate the doses around the usual plasma concentrations of these EDCs (based on previous studies of our lab and other teams) in order to show (or not) a potential non-monotonic dose-response curve
- we added the time in hours in the X axis of figure 1
- we redrew the figure 3
- this notion was already in the introduction of our manuscript but added, as suggested, a point at the beginning of the discussion
- as suggested, we criticized this important point in the discussion (line 255-257)
- we added the references in the discussion regarding the adverse effects of PFOA and PFAS
- we added some references to previous studies in the material section
Kind regards
Prof. Nicolas Chevalier for all the authors
Reviewer 2 Report
The paper “Low doses of PFOA promote prostate and breast cancer cells growth through different pathways” by Charazac et al. properly presents the biological effects of low doses of several POPs (aldrin, BDE28, PFOA, PCB153) that they previously quantified in the circulation of obese patients who underwent bariatric surgery. The authors showed that none of the compounds tested have a cytotoxic effect at doses between micromolar and picomolar concentrations, however, an increase in proliferation was observed at picomolar concentrations of PFOA and PCB153. In addition, the authors performed a mechanistic approach to explain the above-mentioned results. The authors confirmed the results obtained in the screening approach (phosphokinase array and microarray) by additional tests (western blot, RT-qPCR). Moreover, the authors included proper controls in the performed experiments, increasing the value and scientific soundness of the article.
The paper is well structured and well-written, with almost no editing mistakes. However, I have several suggestions for the authors.
1. Line 55-57; try to reformulate “natural history of cancer” doesn’t fit in this context
2. Line 230-231; reformulate the last part of the phrase with as the message is not presented well“but although regulatory agencies”
3. Line 247-252; if you opted to write this paragraph based on “not shown data” try to formulate a hypothesis for the discrepancy observed in the current study and the well-known fact that PFOA interacts with PPARα
Based on these observations, I recommend for this article to be accepted after minor revision.
Author Response
We thank Reviewer #2 for his reviewing and for supporting our manuscript for publication.
As aptly suggested, we reformulated the sentences concerning the cancer history (which was indeed unclear) and the regulatory agencies.
We also added a sentence concerning the interaction between PPAR and PFOA.
Kind regards
Prof. Nicolas Chevalier on behalf of all the coauthors